# State of the Art on Family and Community Health Nursing International Theories, Models and Frameworks: A Scoping Review

**DOI:** 10.3390/healthcare11182578

**Published:** 2023-09-18

**Authors:** Giulia Gasperini, Erika Renzi, Yari Longobucco, Angelo Cianciulli, Annalisa Rosso, Carolina Marzuillo, Corrado De Vito, Paolo Villari, Azzurra Massimi

**Affiliations:** 1Department of Public Health and Infectious Diseases, Sapienza University of Rome, 00185 Rome, Italy; g.gasperini@policlinicoumberto1.it (G.G.); angelo.cianciulli@uniroma1.it (A.C.); carolina.marzuillo@uniroma1.it (C.M.); corrado.devito@uniroma1.it (C.D.V.); paolo.villari@uniroma1.it (P.V.); azzurra.massimi@uniroma1.it (A.M.); 2Department of Translational and Precision Medicine, Umberto I Teaching Hospital, 00161 Rome, Italy; 3Department of Biomedicine and Prevention, University of Rome Tor Vergata, 00133 Rome, Italy; 4Department of Health Sciences, University of Florence, 50134 Firenze, Italy; yari.longobucco@unifi.it; 5Department of Environmental and Prevention Sciences, University of Ferrara, 44121 Ferrara, Italy; annalisa.rosso@unife.it

**Keywords:** family nurse, community nurse, public health nurse, scoping review, WHO

## Abstract

A Family and Community Health Nursing (FCHN) model was first conceptualized by the WHO approximately 25 years ago in response to the epidemiological transition leading to major changes in the population health needs. To date, no study has comprehensively explored the adherence of current applications of FCHN to the WHO original framework. We carried out a scoping review on PubMed, Scopus and CINAHL with the aim to compare the main features of FCHN models developed at the international level with the WHO’s framework. We identified 23 studies: 12 models, six service/program descriptions, four statements and one theoretical model. The FCHN models appear to focus primarily on sick individuals and their family, mainly providing direct care and relying on Interaction, Developmental and Systems Theories. While these features fit the WHO framework, others elements of the original model are poorly represented: the involvement of FCHN in prevention activities is scarce, especially in primary and secondary prevention, and little attention is paid to the health needs of the whole population. In conclusion, current applications of FCHN show a partial adherence to the WHO framework: population approaches should be strengthened in current FCHN models, with a stronger involvement of nurses in primary and secondary prevention.

## 1. Introduction

Chronic diseases and population aging are globally placing an increasing burden on health care systems [1,2,3,4,5], which are required to make an ever-greater commitment to identify and develop strategies to promote healthy lifestyles and increase the early diagnosis and timely treatment of chronic diseases [6]. The COVID-19 pandemic has further highlighted serious shortcomings in the supply of health services, especially for chronic patients. Although elderly and sick people showed a higher risk of serious COVID-19 sequelae [7,8], services for people with chronic diseases have suffered a sharp reduction and sometimes a complete disruption [9,10,11]. Successful strategies have been identified in recent decades for the management of chronic diseases, including the improvement of Primary Health Care (PHC) and the reorientation of health care systems, by reducing hospital centrism and fragmentation of health services [12,13]. At the forefront of this reform are nurses, who, with academic programs that are shaping core competencies and roles in the field of Primary Health Care, are playing a central role in revitalizing PHC and are increasingly becoming leaders in such services [14,15,16].

Over the past 25 years, the World Health Organization (WHO) has outlined the role of nurses in Public Health and PHC by developing specific frameworks for Family and Community Health Nursing (FCHN) [17,18]. The WHO frameworks indicate Family and Community Health Nurses (FCHNs) as a stable focal point for the population at the community level. They directly assist people in their homes or in the community, improving continuity of care, they offer all levels of prevention, with a special focus on addressing social determinants of health, they contribute to policy planning and to resources management, and they support access to and appropriate use of health care services [17,18,19]. It has been highlighted that the definition of a clear and specific nursing model could support professionals in their work, improving the achievement of patient outcomes [20,21] and contributing to strengthening PHC [22]. Furthermore, it is widely recognized that the use of specific conceptual and theoretical frameworks right from the design of educational curricula of nursing advanced programs is essential to protect and preserve attention to the specificity of the contribution of nursing to health care [23,24].

In order to optimize the role of FHCNs in reorienting health systems towards PHC, an assessment of current models of FCHN and their adherence to the WHO framework would be crucial. However, most published studies focus on single models and often do not evaluate their overall adherence to the guiding framework outlined by WHO.

The aim of this work is to provide an overview of current applications of the FCHN model at the international level. A comparison will be made between the main conceptual and organizational components of FHCN models retrieved in literature with those of the WHO framework for Family Nursing, so as to highlight common elements and differences. To allow identification, explanation and condensation of core elements and concepts from a heterogeneous body of knowledge, a scoping review method was chosen [25,26,27].

## 2. Materials and Methods

### 2.1. Study Design

We carried out a scoping review of the literature using the Joanna Briggs Institute methodology [28].

For a transparent reporting, we followed the Preferred Reporting Items for Systematic reviews and Meta-Analyses extension for Scoping Reviews (PRISMA-ScR) Checklist [25], retrieved from the Equator Network (http://www.equator-network.org/, accessed on 4 March 2023) (Appendix A).

### 2.2. Selection Criteria and Search Strategy

The search was performed on PubMed, Scopus, Web of Science and Cumulative Index to Nursing and Allied Health Literature (CINAHL), following database-specific search strategies to collect studies published between 2009 and 2020. The timeframe was chosen in order to identify the most representative models based on the WHO’s framework for Family and Community Health Nursing [17,18]. The search strategy used key terms specifically linked to FCHN combined with those referred to “model” or “framework” (Appendix A). The search was supplemented by scanning the reference lists of the retrieved articles and manual searching. Two independent reviewers removed duplicate articles and screened the title and abstract of all records.

Studies that did not meet the inclusion criteria were excluded. Full texts of potentially relevant articles were examined by two researchers and reasons for exclusion were recorded. Any disagreement was resolved by discussion with a third author. All studies with the following characteristics were included (i) studies aimed to describe theories, conceptual and organizational models and frameworks of FCHN; (ii) studies explaining roles and competencies of FCHNs; (iii) studies focusing on primary care settings; (iv) studies published in English or Italian language, based on co-authors’ language abilities. Articles included specific profiles or specializations in community care such as veterans, military and school nursing, disaster management, FCHN fully focused on a specific type of care (such as stroke care) and educational/academic partnership frameworks were excluded as well as interviews, letters, books and their sections, thesis and conference proceedings.

### 2.3. Data Extraction and Synthesis

For each record included, two reviewers used a standardized data abstraction form to collect the following information: first author, year, country, theory, model, framework, reference theory or model, care receiver, and care setting. Core data related to theory, models and frameworks retrieved, such as theoretical approaches, target population and nurses’ competencies, were matched in a matrix model with the main components of WHO’s Family and Community Health Nursing frameworks [17,18], which are summarized in Table 1. A comprehensive narrative synthesis of the main conceptual and organizational components of retrieved models and of the comparison with the WHO’s framework was produced.

## 3. Results

A total of 4530 records were collected. After duplicates removal, screening of titles and abstracts, 392 studies remained. After full-text assessment, only 23 papers were included in this review [29,30,31,32,33,34,35,36,37,38,39,40,41,42,43,44,45,46,47,48,49,50,51]. The overall selection process is shown in the PRISMA flowchart (Figure 1).

### 3.1. General Results

Among the articles collected, the majority came from the United Kingdom (*n*. 5; 22%) [34,35,37,43,51], the United States of America (*n*. 4; 17%) [29,39,42,50], and Canada (*n*. 3; 13%) [31,36,47]. The other studies were performed in Italy [30,46], The Netherlands [48,49], Australia [38], Ireland [44], Portugal [32], Norway [45], Thailand [40] and Slovenia [41].

Within these articles, we identified twelve models (52%) [32,33,35,38,40,41,42,43,44,46,48,49], six institutional documents about services and programs (26%) [30,31,34,37,45,51], four statements and frameworks of competences (17%) [29,39,47,50] and one theoretical model (5%) of FCHN [36].

The most frequently reported model was the Neighborhood Model [35,43,48,49], described in four of the twelve studies retrieved: two referred to the Buurtzorg model developed in the Netherlands, and the other two to subsequent applications in the UK. The other models retrieved are summarized in Table 2. Services and programs described in literature include the Case Management and Primary Nursing model implemented in Italy [30], the UK Community Matron service [51], the description of a Family Nurse Partners–hip Program in the UK [37] and New Families Program developed in Norway [45]. With regard to competences frameworks, we retrieved the following documents: the American Public Health Association definition and practice of public health nursing [29], the International Family Nursing Association Position Statement on Advanced Practice Competencies for Family Nursing [39], a Competency Framework for Family Practice Registered Nurses developed in Ontario, Canada, following a Deplhi process [47] and the Community/Public Health Nursing Competencies developed by the Quad Council Coalition (alliance of the four US nursing organizations addressing public health nursing issues) [50]. Finally, a study examining the relevance of the theory to the practice of expert public health nurses (PHNs) in Canada was found, which describes the Critical Caring Theory [36].

Care receivers included individuals, families and community members (*n*. 19; 83%) [30,33,34,35,37,38,39,40,41,42,43,44,45,46,47,48,49,50,51]. The main elements of all models were summarized in Table 2.

Geographical distribution of included studies was summarized in Figure 2.

### 3.2. Main Conceptual and Organizational Components of the Models Retrieved

This section analyses the main characteristics of the recovered theories, models and frameworks in the light of a comparison with WHO model (Table 3).

#### 3.2.1. Intervention Provided

FCHNs focus on four main types of interventions: all three levels of prevention and direct care. Direct care is the most frequently reported intervention (*n*. 14; 61%), followed by primary and tertiary prevention (both *n*. 13; 57%), while secondary prevention is the least common nursing intervention (*n*. 6; 26%).

Examples of primary prevention interventions include the provision of educational sessions to improve parent–child relationship or vaccination activities [29,32,33,34,37,38,39,40,41,42,44,45,47,48,49]. Secondary prevention is mainly represented by screening activities, such as cancer and osteoporosis screening [38,39,40,41,44,47,50]. Tertiary prevention includes interventions aimed at reducing the progression of disease, such as improving self-care or medication management [30,34,35,38,39,40,41,43,44,47,48,49,50,51]. Direct care concerns the assessment, planning, realization and evaluation of nursing care activities. Described interventions are various, including, for example, the administration of chemotherapy, wound care, prescription management, emergency and acute care [30,34,35,38,39,40,41,43,44,46,47,48,49,51].

#### 3.2.2. Reference Theory

Most models and frameworks retrieved do not explicitly report a reference theory. Nevertheless, almost all of them defined concepts and praxes that are clearly rooted in the reference theories of the main WHO models [17,18]: Interaction Theory (48%), Systems Theory (35%) and Developmental Theory (17%). Specifically, all elements emphasizing partnership and teamworking between nurse, patient and family were attributed to the Interaction Theory [30,31,33,34,36,37,39,43,45,46,47]; those emphasizing the complexities of factors related to the individual and family system and to the nurse-patient system were attributed to the Systems Theory [29,36,37,38,39,42,45,50]; finally, those emphasizing family development as a non-static system were attributed to the Developmental Theory [33,37,39,45].

In addition, other theories and reference models for FCHN applications were described in the literature, including the Critical Caring Theory [36], the Human Ecology Theory, the Self-Efficacy Theory and Attachment Theory [37], the Health Promotion Model [38,42], the Social Cognitive Theory [38] and Salutogenesis [45]. Finally, the UK guidance document on the development of district nursing specifically refers to the national strategy: “Compassion in Practice: A vision for nurses, midwives and care staff” as reference model [34].

#### 3.2.3. Target Population

FCHN target populations identified by the WHO include individuals, families, members of a specific community (e.g., health needs-based interventions) or general population and communities (e.g., health policy development and implementation). In the included articles, nurses targeting the general population were involved in the planning of health interventions aimed at health issues associated with specific conditions (*n*. 8; 35%), such as homelessness [39,44], or behavioral risk factors (e.g., smoking and sedentary lifestyle) [29,31,34,36,39,42,45,50].

The most represented target population in literature is the family household with its individual members (*n*. 19, 83%), mainly families with elderly and chronically ill patients [30,34,35,36,41,43,44,48,49,51]. In some cases, care recipients were children in the first 1000 days of life and their parents [33,37,45].

#### 3.2.4. Core Competencies

Based on the theories, models and frameworks analyzed, FCHNs are mostly involved in advising and assisting people (*n*. 19; 83%) [30,33,34,35,36,37,38,39,40,41,43,44,45,46,47,48,49,50,51], help coping with health problems (*n*. 16; 70%) [30,33,34,35,36,37,39,40,43,44,45,46,47,48,49,51] and promote health (*n*. 16; 70%) [29,31,34,35,36,37,38,39,40,42,44,45,47,48,49,50]. Nursing activities related disease prevention (prevention of disease, disability and premature death) are less represented in literature (*n*. 11; 48%) [29,31,32,34,37,38,39,42,45,47,50], and so are nursing competencies related to public health, such as population and individual risk assessment (*n*. 8; 35%) [29,31,34,36,38,39,42,50] and interventions on health policies and programs (*n*. 7; 30%) [29,31,32,36,39,40,42,50]. Similarly, competences involving transition of care and collaboration with local professionals, such as pivoting between the family and the primary care physician (*n*. 7; 30%) [35,37,38,41,46,48,49] and promoting early discharge (*n*. 2; 9%) [34,35], were seldom reported.

### 3.3. Comparison between the WHO Matrix and the Identified Frameworks, Models and Theories

Table 3 summarizes the main common features of frameworks, models and theories identified with the WHO frameworks on Family and Community Nursing. The models described in literature are heterogeneous and differ by setting and target populations. Two models have been identified that seem to fit most the original FCHN WHO framework: the International Family Nursing Association (IFNA Position Statement on Advanced Practice Competencies for Family Nursing of) [39] and the Community/Public Health Nursing [C/PHN] Competencies program developed by the Quad Council Coalition (QCC) [50]. The IFNA statement defined a FCHN model oriented to family care, with a comprehensive caretaking perspective: FCHNs should provide interventions aimed at promoting, maintaining, restoring, and strengthening the health of a family and its members, paying attention to community and environmental factors that may influence family health. The FHCN model and competencies are consistent to those defined by the original WHO Family Nurse framework [17]. According to the QCC model, FCHNs should be responsible for the planning of community-oriented health policies and programs, through the assessment of population risks and health needs, and for the delivery of interventions in community settings. The model focuses on disease promotion and prevention, in particular on increasing awareness and adherence to cancer screening programs, and on the self-care of chronic diseases through community engagement, consistent with the WHO model of Community Health Nurses [18]. The other models identified maintain some of the WHO framework’s distinctive features, adapting them to local cultural contexts and target populations. FCHN models mainly include primary, tertiary and direct care prevention activities in specific population subgroups, such as chronic patients, families with newborns and hard-to-reach populations, using a patient-centered care approach, with a focus on specific diseases or health problems.

## 4. Discussion

This scoping review aimed to provide a synthesis of international applications of the Family and Community Health Nursing model, highlighting common elements and differences from the original WHO frameworks for Family and Community Health Nursing. The literature search identified 23 documents, including one theory, 12 models, four frameworks and services/program descriptions and six institutional documents about services and programs, showing a remarkable adherence to the WHO frameworks. Overall, the main characteristics of the identified models include a focus on direct nursing care at the individual and family levels; particularly in community settings, nursing care is involved in health education for chronic diseases, post-acute episodes, and frail populations. Family nurse interventions aimed at enhancing compromised health domains and/or recovery of health status. Common elements among the papers included as a reference theory the adoption of Interaction Theory, which is characterized by the centrality of the relationship with the patient and the interaction between the various systems that impact the family and its health status. The core competencies for their practice are advising and assisting people, helping to cope with health matters and promoting health.

Differently, as appears in the models, theories and frameworks identified, the public health aspects expected by the WHO are less developed: we find limited representation of nursing competencies in community policy planning, as well as in early detection of diseases [52]. The need to strengthen prevention systems, especially primary and secondary prevention, and support approaches aimed at the whole population has been stressed [53,54]. FCHNs are expected to work on prevention activities before a health problem arises and not primarily with ill people and with direct care, as we found in most models [55,56,57]. However, these results are in line with recent literature suggesting that nurses enrolled in FCHN activities are not really confident with this new perspective, remaining anchored to a hospital-based and disease-focused care [58].

Another finding is the discrepancy in the diffusion of the model in non-Western countries, with only one out of the 23 retrieved documents referring to a model developed in Thailand [40]. This may be due to a greater diffusion of FCHN in Western countries, while it seems to be still developing in other areas of the world [59]. Most Western Countries have a long history of Family and Community Nursing, dating back even before with the development of WHO models [60,61,62,63,64]. Our review also highlighted a strong dependence of FCHN models on their context, as shown in the Compassion in Practice Framework [34] and in the Family Nurse Partnership Programme [37], developed in the UK. Both guidance documents appear to be context-sensitive: the first one provides a national vision for community nursing, being very sensitive to the national health care context; the second model shows a very strong focus on the specific target population, represented by first-time young parents. These two examples show how the theoretical references can be adapted according to the context and the objective of the Family and Community Health Nursing model. The ability to adapt Family and Community Health Nursing models according to the context certainly represents a positive factor, which can lead to a nursing practice that is more attentive to the specific needs of a country or context. On the other hand, this could lead to nurses performing roles that are not strictly related to Family and Community nursing and PHC (e.g., emergency room/acute care).

WHO models clearly identify the roots of Family and Community Health Nurses in PHC and in Public Health [17,18]. Both disciplines pay close attention to the whole population, mainly to the healthy population, and to prevention at all levels. However, they also provide a different orientation to nursing practice, as highlighted in the WHO framework for Family Health Nursing [17] and in the next WHO model of Community and Public Health Nursing [18]. While the first framework shows a practice-specific orientation, public health nursing models have a strategic orientation. In this regard, Stanhope and Lancaster defined two types of nursing orientation to people and groups: Community-Oriented Nursing and Community-Based Nursing [65]. The first has been described as a philosophy that permeates Community Health Nursing Practice, focused on the delivery of health care to “community as a whole”, with a positive effect of the “community’s health status (resources)” on people’s health. Community-based nursing has been portrayed as a nursing setting-specific practice, dedicated to sick individuals and their families in their life environment. The nursing practice is “comprehensive, coordinated and continuous” and is centered on both acute and chronic care. Public Health Nurses are therefore focused on primary prevention and health promotion, addressing social determinants of health, and they are rarely involved in direct care. Their practice is focused on the assessment of population needs; on the prevention of disease, disability and premature death, on policies and programs development, planning, implementation and evaluation. On the other hand, FCHNs work is focused on all levels of prevention and on the provision of direct care, while they are seldom involved directly in health promotion activities. They intervene and are competent in supporting people to cope with health matters, advising and assisting, early detecting and treating, acting as a linchpin between the family and the family health physician [33,39,40,44].

Nursing care models show a great variability across different settings, especially in the fields of PHC and Public Health. The scientific literature recognizes the importance and value of nursing theory-guided practice [66]. This approach becomes crucial “for expanding our understanding of the complexity and contexts within which nursing enacts a particular role in the healthcare spectrum” [66,67]. For this reason, the use of a shared conceptual framework, such as those developed by the WHO for Family and Community Health Nursing, would be crucial both to define specific skills of nursing practice, and for the creation of internationally recognized and shared educational curricula [23,24].

### Limitations

This scoping review has several limitations. First, only articles in English and Italian were included, leading to a possible selection bias of articles. Second, as a scoping review, this article does not offer an appraisal of methodological biases of the literature included. However, the aim of this study was not to provide a narrative nor quantitative synthesis of evidence, but to pool together, summarize and identify key elements about a various body of knowledge [25,26,27]. Finally, only articles published between 2009 and 2020 were searched, limiting the findings to recent theories, models and framework; on the other hand, this timeframe allowed identifying already consolidated models, while also excluding potential distortions caused by the COVID-19 pandemic.

## 5. Conclusions

Even 25 years after their initial conceptualization and despite multiple influences, international FCHN theories, models and frameworks are not far from the original WHO models that guided their development. Despite the general good level of adherence to the WHO frameworks for FCHN, current models need to be reoriented towards a stronger focus on population and prevention, especially primary and secondary prevention.

## Figures and Tables

**Figure 1 healthcare-11-02578-f001:**
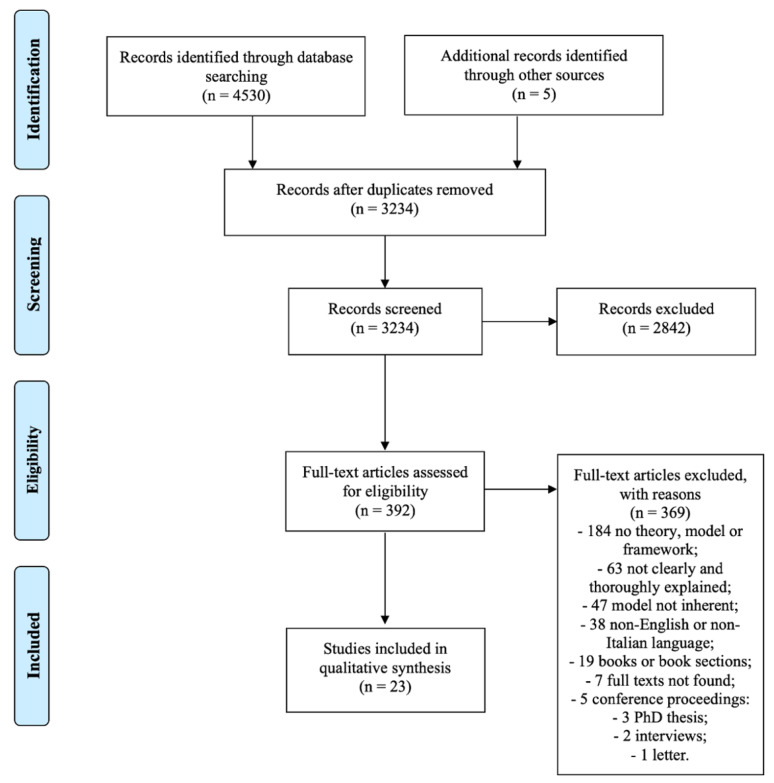
PRISMA flow diagram of the review process.

**Figure 2 healthcare-11-02578-f002:**
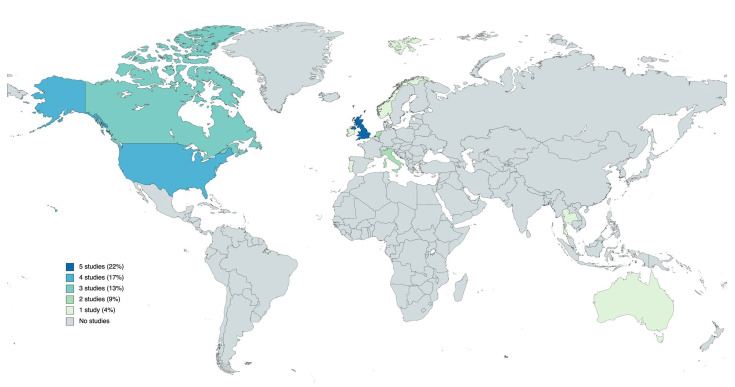
Geographical distribution of included studies.

**Table 1 healthcare-11-02578-t001:** Main components of the WHO matrix framework.

	Type of Interventions	Reference Theory Model	Target Population	Core Competencies
WHO Family and Community Health Nursing frameworks [17,18]	Primary prevention Secondary prevention Tertiary prevention Direct care	Systems Theory Interaction Theory Developmental Theory	Individuals, families, community members Communities, populations	Help to cope with health matters Advise and assist Early detect and treat Facilitate early discharge Act as the lynchpin between the family and the family health physician Assessment of population needs Health promotion Focus on social determinants of health Prevention of disease, disability and premature death Policy and program development, planning, implementation, evaluation and advocacy

**Table 2 healthcare-11-02578-t002:** General characteristics of included studies.

First Author, Year	Country	Theory, Model, Framework	Reference Theory or Model	Care Receiver	Care Setting
American Public Health Association, Public Health Nursing Section, 2013 [29]	United States of America	The definition and practice of public health nursing	Systems Theory	Entire population, including sub-populations; individuals, families, communities and the systems.	Not reported
Bargna E., 2011 [30]	Italy	Case Management (Community Matrons’ approach)	Interaction Theory	Patients with medium–low complexity, cancer; multi-pathological elderly.	Home; Community; Nurse Clinic
		Case Management (or Second Level Primary Nursing)		Chronic and terminal multi-pathological elderly.	Home
		Primary Nursing		Chronic and terminal multi-pathological elderly; people with medium–low complexity.	Home; Nurse Clinic
Cusack C., 2017 [31]	Canada	Professional practice model to promote population health and equity.	Interaction Theory	Communities and populations.	Not reported
Da Cunha C., 2020 [32]	Portugal	Specialist nurse in community and public health nursing	Not reported	Populations, communities and groups.	Not reported
Day C., 2013 [33]	Not reported	Family Partnership Model	Interaction Theory; Developmental Theory	Child and family. More recently, it has been used in the fields of adult mental health and learning disability.	Not reported
Department of Health Public health nursing, 2013 [34]	United Kingdom	The District Nursing Service Model	Compassion in Practice Framework; Interaction Theory	Population and ill ones.	Community settings; Home
Downes C., 2009 [35]	United Kingdom	Neighborhood Model	Not reported	Patients with complex long-term physical and mental conditions.	Home; respite setting
Falk-Rafael A., 2012 [36]	Canada	Critical Caring mid-range Theory	Critical Caring Theory; Interaction Theory; Systems Theory	Individuals, families, community members, communities and populations.	Not reported
Family Nurse Partnership National Unit, 2012 [37]	United Kingdom	The Family Nurse Partnership Program	Human Ecology Theory, Self-Efficacy Theory; Attachment Theory; Interaction Theory; Developmental Theory; Systems Theory	Vulnerable young first-time mothers.	Home
Hungerford C., 2016 [38]	Australia	Community-based, clinic-located Nurse Practitioner model of practice	Social Cognitive Theory; Health Promotion Model; System Theory	Local residents of the tourist center and tourists who stay in the area short term.	Community-based clinic in a remote tourist destination where there is no resident General Practitioner; home
International Family Nursing Association, 2017 [39]	United States of America	Advanced Practice Competencies for Family Nursing	Systems Theory; Interaction Theory; Developmental Theory	Diverse families and individuals in all types of health care conditions and settings.	Not reported
Jongudomkarn D., 2014 [40]	Thailand	The Khon Kaen University Family Health Nursing & The Memimema Model	Not reported	Individuals, families and community members.	Primary Health Care Unit
Klemenc-Ketis Z., 2019 [41]	Slovenia	Model of Comprehensive Care	Not reported	People with difficulties accessing health care; with or without any risk factors and those with the most common chronic disease.	Home
Kulbok P.A., 2012 [42]	United States of America	Community Participatory Health Promotion Model	Community Participatory Health Promotion Model; Systems Theory	High-risk, vulnerable populations, frail elderly, homeless individuals, sedentary individuals, smokers, teen mothers, and those at risk for a specific disease.	Diverse community settings, for instance: home health agencies
Lalani M., 2019 [43]	United Kingdom	Buurtzorg Model	Interaction Theory	Local service user with complex health and social problems.	General Practice surgery; home
Leahy-Warren P., 2017 [44]	Ireland	Nursing and Midwifery in the Community Model	Not reported	Individual, family or community from conception to death, and from health to chronic illness.	Not reported
Leirbakk M.J., 2019 [45]	Norway	New Families Program	Salutogenesis; Interaction Theory; Developmental Theory; Systems Theory	First-time families and their infants.	Home; clinic.
Marcadelli S., 2019 [46]	Italy	Not reported	Interaction Theory	Mainly chronically ill and older people.	Nurses’ own offices, in General Practitioners’ surgeries or in other locations
Moaveni A., 2010 [47]	Canada	Competency Framework for Family Practice Registered Nurses	Interaction Theory	Individuals, families, community members.	Not reported
Monsen K., 2013 [48]	Netherland	Buurtzorg Model (Dutch for “neighborhood care”)	Not reported	Elderly, disabled, patients in need of home, hospice, and dementia care.	Home and Office
Monsen K., 2013 [49]	Netherland	Buurtzorg Model	Not reported	Elderly with disabilities, terminal illness, with chronicity and/or dementia.	Home
Quad Council Coalition Competency Review Task Force, 2018 [50]	United States of America	Community/Public Health Nursing Competencies	Systems Theory	Population, communities and their members, families and individuals.	Not reported
Young J., 2010 [51]	United Kingdom	Community Matron service	Not reported	Elderly with multiple long-term conditions (considered at exacerbation risk).	Home

**Table 3 healthcare-11-02578-t003:** Influences retrieved linked to the WHO frameworks.

Family and Community Health Nurse Models										
Author, Year	Type of Interventions				Reference Theory Model			Target Population		Core Competencies
	Primary Prevention	Secondary Prevention	Tertiary Prevention	Direct Care	Systems Theory	Interaction Theory	Developmental Theory	Individuals, Families, Community Members	Communities, Populations	1–10 *
APHA, 2013 [29]	X				X				X	6; 7; 8; 9; 10;
Bargna E., 2011 [30]			X	X		X		X		1; 2; 3;
Cusack C., 2017 [31]	X					X			X	6; 7; 8; 9; 10;
Da Cunha C., 2020 [32]	X								X	9; 10;
Day C., 2013 [33]						X	X	X		1; 2;
Department of Health Public health nursing, 2013 [34]	X		X	X		X		X	X	1; 2; 4; 6; 7; 8; 9;
Downes C., 2009 [35]			X	X				X		1; 2; 3; 4; 5; 7;
Falk-Rafael A., 2012 [36]					X	X		X	X	1; 2; 6; 7; 10;
Family Nurse Partnership National Unit, 2012 [37]	X				X	X	X	X		1; 2; 5; 7; 9;
Hungerford C., 2016 [38]	X	X	X	X	X			X		2; 5; 6; 7; 9;
IFNA, 2017 [39]	X	X	X	X	X	X	X	X		1; 2; 6; 7; 9; 10;
Jongudomkarn D., 2014 [40]	X	X	X	X				X		1; 2; 3; 7; 10;
Klemenc-Ketis Z., 2019 [41]	X	X	X	X				X		2; 5;
Kulbok P.A., 2012 [42]	X				X				X	6; 7; 8; 9; 10;
Lalani M., 2019 [43]			X	X		X		X		1; 2;
Leahy-Warren P., 2017 [44]	X	X	X	X				X		1; 2; 7; 8;
Leirbakk M.J., 2019 [45]	X				X	X	X	X	X	1; 2; 3; 7; 9;
Marcadelli S., 2019 [46]				X		X		X		1; 2; 3; 5;
Moaveni A., 2010 [47]	X	X	X	X		X		X		1; 2; 7; 8; 9;
Monsen K., 2013 ** [48,49]			X	X				X		1; 2; 5; 7;
Quad Council Coalition Competency Review Task Force, 2018 [50]	X	X	X		X			X	X	2; 6; 7; 8; 9; 10;
Young J., 2010 [51]			X	X				X		1; 2; 3;

* 1. Help to cope with health maters; 2. advise and assist; 3. early detect and treat; 4. facilitate early discharge; 5. act as the lynchpin between the family and the family health physician; 6. assessment of population needs 7. health promotion; 8. focus on social determinants of health; 9. prevention of disease, disability and premature death; 10. policy and program development, planning, implementation, evaluation and advocacy. ** Represent both articles by Monsen K. et al. published in 2013.

## Data Availability

Not applicable for the methodology of this study.

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
