# Peer review of "State of the Art on Family and Community Health Nursing International Theories, Models and Frameworks: A Scoping Review"

_healthcare, 2023, doi:10.3390/healthcare11182578_

Round 1

Reviewer 1 Report

On the whole, I found the article to be well-crafted and insightful. However, I do have a few comments and suggestions for enhancement.

Materials and Methods

1. The name of the figure is situated beneath the illustration (Figure 1)

2. There seems to have been an issue with the manuscript template as the line numbers are not appearing in the results section.

3. In the "General Results" section, it might be more effective to present the quantity and percentage distribution across countries as a table in subsection 3.1.

4. In the sentence “Care receivers included individuals, families and community members (n. 19; 83%) [30, 33 – 38, 39 – 41, 43 – 47, 48, 49 – 51]. The main elements of all models were summarized in Table” table number not specified.

Results

5. It is recommended that the Prisma flow chart be placed inside of the methods section.

6. Could you kindly provide information on how the duplicate articles were eliminated or excluded from the analysis?

7. What other literature source has been used for additional literature search?

8. Didn't it be apparent at the level of titles and abstracts whether the full-text articles are books or PhD theses during the selection process?

9. It could be considered advantageous to consolidate the data presented in Table 2 based on both the year and the country. Given that the percentage for the UK appears to be relatively higher compared to other countries, placing it at the forefront might be a logical arrangement. Alternatively, if the data is being organized chronologically by years, following that sequence would be equally coherent.

10. I would like to bring to your attention that the title of Table 2 does not accurately convey its content or purpose.

11. I kindly request clarification regarding the inclusion of Cunha 2020 (31) in the general results, as it appears that there is no reference to a theory or theoretical framework, as indicated in Table 2.

12. I appreciate your feedback regarding the referencing of tables in the text. Your suggestion to maintain a consistent referencing style from various sources is duly noted.

13. May I inquire about the statistical software you used to process the data for the general results?

References

14. Could you please review the reference list? There appear many sources that are older than five years

Author Response

Dear Reviewer,

thank you for your comments and suggestions, which were very helpful in revising the manuscript.

Below are the changes made according to your comments.

-------

On the whole, I found the article to be well-crafted and insightful. However, I do have a few comments and suggestions for enhancement.

  1. The name of the figure is situated beneath the illustration (Figure 1)

We thank the Reviewer for this comment. We placed description of Figure 1 in the correct location (page 5, line 6).

  1. There seems to have been an issue with the manuscript template as the line numbers are not appearing in the results section.

We thank the Reviewer for this comment. We restored the page and line numbers.

  1. In the "General Results" section, it might be more effective to present the quantity and percentage distribution across countries as a table in subsection 3.1.

We thank the Reviewer for this comment. We realized a chart to increase the understanding of the geographic distribution of the included studies. The new figure has been included in the General Results section (page 10).

  1. In the sentence “Care receivers included individuals, families and community members (n. 19; 83%) [30, 33-41, 43-51]. The main elements of all models were summarized in Table” table number not specified.

We thank the Reviewer for this comment. We have corrected the typo.

  1. It is recommended that the Prisma flow chart be placed inside of the methods section.

Thank you very much for the suggestion, however the PRISMA checklist for Scoping reviews (http://www.prisma-statement.org/Extensions/ScopingReviews), requires that the flowchart be included in the results section. For the purpose of complying with PRISMA guidelines, we chose to leave the flowchart in the results.

  1. Could you kindly provide information on how the duplicate articles were eliminated or excluded from the analysis?

We thank the Reviewer for this comment. We specified in the methods that the duplicate removal process and screening by title and abstract was done by two independent reviewers through Zotero Citation Manager.

Methods, page 2, lines 90 – 91: “Two independent reviewers removed duplicate articles and screened the title and abstract of all records”.

  1. What other literature source has been used for additional literature search?

Articles identified through other sources were found by scanning the reference lists of the retrieved articles and manual searching. We listed additional resources in the methods section (page 2, lines 89 – 90).

  1. Didn't it be apparent at the level of titles and abstracts whether the full-text articles are books or PhD theses during the selection process?

Thank you for the question. For PhD theses and books entered as non-eligible for full texts, excluding them was not possible just from reading the titles and abstracts.

 9. It could be considered advantageous to consolidate the data presented in Table 2 based on both the year and the country. Given that the percentage for the UK appears to be relatively higher compared to other countries, placing it at the forefront might be a logical arrangement. Alternatively, if the data is being organized chronologically by years, following that sequence would be equally coherent.

Thank you for the suggestion, we agree that grouping results by year in Table 2 may be more informative. However, we decided to keep the presentation of the included articles in alphabetical order in the table, this is so as not to change the bibliography in Vancouver. It is, however, possible to distinguish the geographic distribution in the new Figure 2 and in the subsection "3.1 General Result."

  1. I would like to bring to your attention that the title of Table 2 does not accurately convey its content or purpose.

Thank you for the suggestion. We re-titled the table as follows: "Table 2. General characteristics of included studies."

  1. I kindly request clarification regarding the inclusion of Cunha 2020 in the general results, as it appears that there is no reference to a theory or theoretical framework, as indicated in Table 2.

Thank you for the suggestion. We have corrected the omission. We have included the model described by Da Cunha 2020 in Table 2 (page 7).

  1. I appreciate your feedback regarding the referencing of tables in the text. Your suggestion to maintain a consistent referencing style from various sources is duly noted.

We thank the Reviewer for this comment.

  1. May I inquire about the statistical software you used to process the data for the general results?

We thank the Reviewer for this comment. Data were presented through a narrative synthesis; statistical software was not used for data processing. Frequencies and percentages were calculated by the authors.

  1. Could you please review the reference list? There appear many sources that are older than five years.

We thank the Reviewer for this comment. We have updated the bibliography. Updated references are highlighted in red in the manuscript. References describing official documents and/or historical processes could not be updated, therefore the bibliography still maintains 5-year-old non-updated references.

Reviewer 2 Report

This is an interesting and topical piece of work that needs minor revision though. I will cite a number of amendments that the authors need to consider before resubmitting the manuscript.

1. A new paragraph should start from this point: "Over the past 25 years, 48 the World Health Organization (WHO)". Diving paragraphs has been an issue elsewhere, e.g. Discussion. Please revise accordingly.

2. "Health determinants" is not a sound term. It should be replaced as follows: https://www.who.int/health-topics/social-determinants-of-health

3. A more general observation: Why school nurses are not included in this manuscript as part of the wide community nursing work? This has to be explained in the text.

4. The Discussion section needs greater revision. It needs to adopt a more narrative approach, to describe and explain to the unfamiliar reader what are the key findings of this scoping review and what are the principal conclusions. I would advise to avoid overusing acronyms that impede the flow of the text. Last but not least the authors need to discuss their findings in relation to the literature, which is weak in the current manuscript. 

The English is good  but the Discussion lacks clarity.

Author Response

Dear Reviewer,

thank you for your comments and suggestions, which were very helpful in revising the manuscript.

Below are the changes made according to your comments.

-------

  1. A new paragraph should start from this point: "Over the past 25 years, the World Health Organization (WHO)". Diving paragraphs has been an issue elsewhere, e.g., Discussion. Please revise accordingly.

We thank the Reviewer for this comment. We revised the sections according to the suggestions.

  1. "Health determinants" is not a sound term. It should be replaced as follows: https://www.who.int/health-topics/social-determinants-of-health

We thank the Reviewer for this comment. We revised the term as suggested by the WHO. The term is uniformed in the text as follows: “social determinants of health” (page 2, lines 55; page 16, line 60).

  1. A more general observation: Why school nurses are not included in this manuscript as part of the wide community nursing work? This has to be explained in the text.

We thank the Reviewer for this comment. We excluded Family and Community Health Nurses who held specialist roles within the community such as school nurses, disaster managers etc. We included in the methods section an explanation regarding the choice to exclude specialized figures.

“Articles included specific profiles or specializations in community care such as veterans, military and school nursing, disaster management, FCHN fully focused on a specific type of care (such as stroke care) and educational/academic partnership frameworks were excluded as well as interviews, letters, books and their sections, thesis and conference proceedings” (page 3, lines 98 - 100).

  1. The Discussion section needs greater revision. It needs to adopt a more narrative approach, to describe and explain to the unfamiliar reader what are the key findings of this scoping review and what are the principal conclusions. I would advise to avoid overusing acronyms that impede the flow of the text. Last but not least the authors need to discuss their findings in relation to the literature, which is weak in the current manuscript. 

We thank the reviewer for this comment. We revised the discussion. The revised sections are highlighted in red in the manuscript (page 15, lines 7 - 20). The description of the main results has changed, aiming to give a more organic and narrative view of the main findings. The first paragraph discusses the common features among the models and the main profile of FCHN found; the second paragraph summarizes the main differences compared with the WHO framework. Finally, some relevant findings were discussed, including the geographical distribution and difference between family-, community- and public health-oriented models. Lastly, we eliminated the more technical sections (e.g., reference theories) and reduced the use of acronyms.

Reviewer 3 Report

Thank you for submitting this scoping review on the application of the FCHN WHO framework to models of nursing practice. I appreciate the international scope of the article and how this framework is being applied in different countries. Sometimes the reader can get confused with the terminology. For example, is the WHO FCHN work a distinct model in itself or a framework that others have constructed models from? It may help to provide some conceptual clarity by presenting definitions of what constitutes a theory, a framework, and a model. For example, are you using the terms "framework" and "model" synonymously?

Can you explain why your review of articles began in 2009 and not earlier or later? I understand why you stopped in 2020 (because of COVID). However, it may be that, during COVID, there was an expansion of these models when the field of public health was thrust into the forefront of the public conscience.  Do you have additional recommendations for further research? Where do you plan to take your research following the scoping review? 

Finally, I might suggest you shorten the title. The second part of the title would be sufficient to describe the nature of your research.

The manuscript could benefit from additional editing but is certainly readable and understandable.

Author Response

Dear Reviewer,

thank you for your comments and suggestions, which were very helpful in revising the manuscript.

Below are the changes made according to your comments.

--------------

  1. Thank you for submitting this scoping review on the application of the FCHN WHO framework to models of nursing practice. I appreciate the international scope of the article and how this framework is being applied in different countries. Sometimes the reader can get confused with the terminology. For example, is the WHO FCHN work a distinct model in itself or a framework that others have constructed models from? It may help to provide some conceptual clarity by presenting definitions of what constitutes a theory, a framework, and a model. For example, are you using the terms "framework" and "model" synonymously?

We thank the Reviewer for this comment. We standardized terminology in the manuscript. For the purpose of the scoping review, international implemented models or frameworks for FCHN were identified. In the text, "models" or "frameworks" were discriminated through the general results and Table 2 according to the instructions provided by the authors. Otherwise, we standardized terminology by using the term "framework" for the WHO documents used for comparison (17, 18). As described in the manuscript the WHO framework is the first document defining the role of the FCHN and was used to compare different definitions of the role of the FCHN at the international level. The other models and frameworks identified are derived from the WHO's first definition of the FCHN.

  1. Can you explain why your review of articles began in 2009 and not earlier or later? I understand why you stopped in 2020 (because of COVID). However, it may be that, during COVID, there was an expansion of these models when the field of public health was thrust into the forefront of the public conscience.  Do you have additional recommendations for further research? Where do you plan to take your research following the scoping review? 

We thank the Reviewer for this comment. Thank you for the question. We defined this time frame using the pandemic from COVID-19 as a starting point, which highlighted the relevance of nurses working in community settings as FCHNs. We excluded articles published during the pandemic because the organizational models described were affected by the reorganization of community-based services and the pandemic. Our intent was to define FCHN models implemented in "ordinary" community settings, and for this reason we started from 2020 and investigated the literature of the last 10 years to identify the most established and most recent models. However, it is our intention to conduct a new systematic review to identify new organizational models implemented during COVID-19 and compare them with all models published since 2000 (WHO framework).

  1. Finally, I might suggest you shorten the title. The second part of the title would be sufficient to describe the nature of your research.

We thank the Reviewer for this comment. We revised the title as suggested.

Round 2

Reviewer 1 Report

No more suggestion.

Reviewer 3 Report

Thank you for your thoughtful response to the reviewers comments and for clarifying the items that were pointed out by the reviewers. I am comfortable with the revisions that have been made and am pleased to recommend your manuscript for publication.

The manuscript would benefit from editorial revision but overall it is clear and well organized.